# Drug Utilization of Rifaximin-α in Patients with Hepatic Encephalopathy: Evidence from Real Clinical Practice in Italy

**DOI:** 10.3390/medicina61020221

**Published:** 2025-01-26

**Authors:** Valentina Perrone, Marcello Usala, Chiara Veronesi, Maria Cappuccilli, Luca Degli Esposti

**Affiliations:** 1CliCon S.r.l. Società Benefit, Health, Economics & Outcomes Research, Via Murri 9, 40137 Bologna, Italy; chiara.veronesi@clicon.it (C.V.); maria.cappuccilli@clicon.it (M.C.); luca.degliesposti@clicon.it (L.D.E.); 2Alfasigma S.p.A., 40133 Bologna, Italy; marcello.usala@alfasigma.it

**Keywords:** dispensing modality, drug utilization, hepatic encephalopathy, rifaximin-α

## Abstract

*Background and Objectives*: This analysis described rifaximin utilization in Italian patients with hepatic encephalopathy (HE). Although rifaximin is effective in preventing HE relapses, therapeutic management and prescriptive attitudes might be improved. *Materials and Methods*: Between Oct-2020 and Sep-2021, approximately 12.7 million citizens, patients hospitalized for HE, were identified through the ICD-9-CM code 572.2. Among those discharged alive, utilization of rifaximin 550 mg vs. rifaximin 200 mg for two months post-admission was compared. *Results*: Of 634 patients hospitalized for HE, 447 (70.5%) were discharged alive. In the two following months, 276 (61.7%) received rifaximin, of whom 117 (26.2%) received rifaximin 550 mg (two daily tablets) and 159 (35.6%) received rifaximin 200 mg (six daily tablets); among 171 patients without rifaximin, 56 (32.7%) received lactulose/lactitol. One year after rifaximin initiation, patients on rifaximin 550 mg (vs. 200 mg) were more frequently persistent (i.e., did not interrupt therapy) (78.6% vs. 46.9%, *p* < 0.001), showed a lower switching proportion (21.4% vs. 40.7%, *p* < 0.050), and had a mean monthly dose closer to the recommendations of 36,000 mg/month (~33,000 mg/month vs. 11,629 mg/month, respectively). *Conclusions*: This analysis suggests suboptimal rifaximin utilization for HE management. Although rifaximin 550 mg is the only formulation with specific indication and reimbursability to prevent HE relapses in Italy, rifaximin 200 mg is more largely used. The need to improve rifaximin prescribing choices is supported by higher persistence, lower switching rates, and average doses aligned to recommendations in patients treated with rifaximin 550 mg.

## 1. Introduction

Hepatic encephalopathy (HE) is a severe and disabling complication of cirrhosis, characterized by brain dysfunction due to the accumulation of pro-inflammatory cytokines and gut-derived neurotoxins, in particular ammonia [1,2]. HE is associated with a worse quality of life, decreased survival, and a considerable risk of recurrence for patients. Neurological impairment can be reversible only if HE is diagnosed early and properly treated [1,2,3,4].

Among the treatment options for HE, rifaximin-α (RFX) has shown effectiveness in the prevention of recurrent episodes of HE in adults [3,4]. RFX is an oral, non-absorbable, gut-targeted antibiotic derived from rifamycin. RFX has bacteriostatic action and a broad antimicrobial spectrum, including ammonia-producing bacteria with a key role in HE pathogenesis [5].

The use of RFX in combination with lactulose has been recommended by the latest guidelines of the European Association for the Study of the Liver (EASL) as secondary prophylaxis after more than one additional episode of overt HE within 6 months of the first one [6]. There is much evidence to show that RFX, in combination with a non-absorbable disaccharide, can reduce overall mortality risk, the likelihood of incurring serious adverse events, hospitalization length, and the occurrence/recurrence of HE [7,8]. RFX is available in two formulations of 550 mg and 200 mg, with several differences concerning indications [4,9], prescription and dispensing modality, recommended dosage, and summary of product characteristics.

Since 2016, RFX 550 mg twice daily has been specifically authorized and reimbursed in Italy for reducing the risk of overt HE recurrence in adult patients [10]. In the past, RFX 200 mg formulations administered with a total dose of 1200 mg/day (six daily tables) were used for maintaining HE remission [11], but following the availability of RFX 550 mg, recent guidelines recommend this latter formulation (two daily tables to reach a total dose of 1100 mg RFX) for the same purpose [8]. According to the determinations of the Italian Medicines Agency (AIFA), RFX 200 mg is currently indicated for other conditions than HE, specifically acute and chronic intestinal infections sustained by Gram-positive and Gram-negative bacteria, diarrheal syndromes, diarrhea caused by the altered balance of intestinal microbial flora, pre- and post-operative prophylaxis of infectious complications in gastrointestinal tract surgery, and adjuvant in the therapy of hyperammonemia; for this latter indication, the recommended daily dosage is of 1200 mg/day (six daily tables) [12,13].

It is known that increased pill burden is among the barriers to medication adherence [14,15], and this poses the question of whether the use of RFX 550 mg might facilitate the therapeutic management of HE from patients’ perspectives. Moreover, despite the evidence-based effectiveness of RFX at the dosage of 1100 mg as recommended by the guidelines, there are still some gaps in the HE setting related to prescribing behaviors [14]. Hence, the possible impact of RFX formulation (200 mg or 550 mg) for the treatment of HE patients on drug utilization emerges as a central issue.

In this framework, the mode of delivery of the available formulations of RFX might be among the main determinants of therapeutic compliance in terms of adherence and persistence. In Italy, drugs reimbursed by the Italian National Healthcare System (INHS) are provided to patients through two main distribution models, hospitals and community pharmacies. In the “centralized” approach called “direct distribution” (DD), drug provision occurs directly through the hospital channel. The other model is based on drug distribution through territorial pharmacies on behalf of hospitals, commonly defined in Italy as “on-behalf distribution” (“distribuzione per conto” or DPC). Both models have pros and cons. The DD channel allows cost savings from the INHS perspective but might imply higher expenses for patients due to organizational aspects (e.g., travel costs). On the other hand, the DPC model offers a more capillary supply of drugs and might facilitate access to medications for patients and caregivers [16].

The formulation RFX 550 mg belongs to medicinal products subject to restrictive medical prescriptions, so it can be delivered to the patients on prescription by hospitals or specialists (gastroenterologists, infective disease specialists, or internists) [10]. The type of distribution, DD or DPC, is established on a regional basis: in some regions, the drug is exclusively dispensed through DD, while other regions use both DD and DPC routes. The formulation RFX 200 mg belongs to the class of reimbursable essential medications and drugs for chronic diseases that can be prescribed by the general practitioner (GP) and are dispensed by community pharmacies.

A summary of the main differences between the two formulations of RFX is reported in Table 1.

The present analysis was undertaken to provide an up-to-date snapshot of the management of HE patients in a setting of real clinical practice in Italy. The primary aim was to investigate whether the type of RFX formulation (550 mg or 200 mg) prescribed to patients with HE and dispensing modality (DD and DPC) might influence drug utilization.

## 2. Materials and Methods

### 2.1. Data Source

A retrospective observational analysis was conducted using data extrapolated from the administrative flows of a pool of Italian healthcare entities (Local Health Units, LHUs) covering approximately 12.7 million health-assisted citizens and data availability from January 2009 to December 2023. These entities were selected by their geographical distribution (by North, Center, and South Italy), by data completeness, and by the high-quality linked datasets. Administrative databases are large data repositories on healthcare services/resources delivered and reimbursed by the INHS. The following databases were used: (i) beneficiaries’ database for patients’ demographics; (ii) pharmaceutical database for data on drug prescriptions with their Anatomical Therapeutic Chemical (ATC) code; (iii) hospitalization database for hospital discharge diagnoses classified by the International Classification of Diseases, Ninth Revision, Clinical Modification (ICD-9-CM); (iv) exemption database for active payment waiver codes associated with specific disease diagnoses; and (v) outpatient specialist service (OSS) database for data on specialist visits, diagnostic procedures, and laboratory tests.

Approval has been obtained from the ethics committees of the involved local health units. The dataset used consists solely of anonymized data.

### 2.2. Study Design and Selection Criteria

From October 2020 to September 2021 (enrollment period), adult patients (>18 years) hospitalized for HE were included using as a diagnosis proxy the presence of at least one hospitalization discharge diagnosis (primary or secondary) with the ICD-9-CM code 572.2 (hepatic coma). The date of the first hospitalization for HE during the enrollment period was defined as the index date. Only patients discharged alive were considered for the analysis. The characterization period was the whole period of data availability (at least 1 year) before the index date, and the follow-up was all the available period after the index date. Exclusion criteria were age < 18 years and no continuous data availability within the databases during the study period (for instance, patients who moved to another region).

### 2.3. Treatments in Analysis

After inclusion, patients were defined as treated with at least one prescription of RFX during the first 2 months after the index date (the time of HE-related hospital discharge). Treated patients were stratified according to the type of treatment schedule with RFX started in the first 2 months after the index date (hospital discharge): RFX 550 mg and RFX 200 mg. Both formulations share the same ATC code (A07AA11). RFX 550 mg (misnan code: 041924046) is available in packages of 56 film-coated tablets, with a recommended dose of 2 tablets/day covering 28 days of therapy, while RFX 200 mg (misnan codes: 025300029, 025300025) is available in packages of 12 film-coated tablets, with a recommended dose of 6 tablets/day covering 2 days of therapy.

The number and proportion of patients on treatments other than RFX, namely lactulose (ATC code: A06AD11o or V03AB with minsan codes: 022711129, 023535166, 024409144) and lactitol (ATC code: A06AD12 or V03AB with minsan codes: 029563018, 029563044), were also examined.

### 2.4. Patients’ Baseline Characteristics

For all the patients included, the demographic characteristics were collected at the index date, namely age and gender distribution expressed as the percentage of male subjects. During the characterization period, patients’ clinical history was investigated, searching for previous cirrhosis-related complications, namely ascites, portal hypertension, liver primary malignant tumors, alcohol dependence syndrome, and extra-hepatic comorbidities, namely diabetes, chronic kidney disease, cardiovascular disease, cerebrovascular disease, cancer, and psychiatric disorders. These conditions were identified by means of hospitalization discharge diagnoses, drug prescriptions, and/or exemption codes as proxies of disease diagnosis.

### 2.5. Drug Utilization

Treatment interruption was defined by the absence of the index prescription during the last three months of the follow-up period; conversely, persistence was defined by the presence of the index prescription during the last three months of the follow-up period.

Switching was defined as the change from the first medication prescribed to a different one, including a change in dose form (e.g., RFX 200 mg to RFX 550 mg and vice versa).

Alternate treatment was defined as rotation between RFX 200 mg and RFX 550 mg.

Average monthly dosage was the sum of milligrams contained in each package of RFX 200 mg and RFX 550 mg, dispensed during the first year of follow-up, divided by 12 (deaths and switches excluded).

A focused analysis was conducted on patients enrolled using RFX 550 mg prescription as a unique inclusion criterion between October 2020 and September 2021. For this subgroup, the date of the first prescription of RFX 550 mg was considered the index date; the characterization period was all the time interval available (at least 12 months) prior to the index date, and the follow-up was the 12-month period after the index date. The patients were defined as “naïve” if they had not received any prescription of RFX 550 mg before the index date or as “experienced” in the presence of a previous prescription of RFX 550 mg during the 120 days before the index date.

Given that in Italy, RFX 550 mg has two different modes of distribution, which can be either in-hospital (DD) or via territorial pharmacies (DPC), depending on the region, the potential rebounds of the different dispensing modalities on the management of HE patients and drug utilization were also investigated. For this purpose, patients were stratified by regions with both DPC and DD channels for RFX 550 mg distribution and regions with DD only.

### 2.6. Statistical Analysis

Continuous variables are reported as the mean ± standard deviation (SD), and categorical variables as frequencies and percentages. Continuous variables were analyzed by Student’s t-test and categorical variables by chi-square test. A multivariable logistic regression model adjusted for confounders was developed to calculate odds ratios (OR) with their corresponding 95% confidence interval (95%CI) for the variables of interest potentially influencing persistence on therapy, including age, gender, concomitant hepatic, cardiovascular, cerebrovascular, and psychiatric conditions, and type of RFX formulation at starting therapy. Model calibration was assessed using the Hosmer–Lemeshow goodness-of-fit test.

A *p*-value < 0.05 was considered statistically significant, and all the analyses were performed using Stata SE version 17.0 (StataCorp, College Station, TX, USA).

## 3. Results

As shown in Figure 1, among 634 HE patients identified, 447 (70.5%) were discharged alive. In the two months following index hospitalization, 99 out of 447 (22.1%) died, 171 (38.3%) did not receive RFX, and 276 (61.7%) received RFX, of whom 117 (26.2%) received RFX 550 mg and 159 (35.6%) received RFX 200 mg. Among the 171 patients without RFX therapy, 42 (24.6%) were treated with lactulose and 14 (8.2%) with lactitol, while the remaining ones were untreated.

The baseline demographic and clinical characteristics of patients stratified by type of starting RFX treatment are detailed in Table 2. Among 117 patients treated with RFX 550 mg and 159 treated with RFX 200 mg, there were no significant differences in terms of mean age (respectively, 65.9 years and 67.3 years, *p* = 0.315) and gender distribution (males: 55.6% and 65.4%, *p* = 0.097). The rate of cirrhosis complications for the RFX 550 mg vs. RFX 200 mg groups was also quite similar: ascites was found, respectively, in 43.6% vs. 34.6% (*p* = 0.129), portal hypertension in 19.7% vs. 17.6% (*p* = 0.665), and liver tumor in 9.4% vs. 8.8% (*p* = 0.864). Considering extra-hepatic comorbidities, diabetes was the most frequently detected (about 49–50% in both groups), followed by cancer (20.5% vs. 25.8% in patients with RFX 550 mg and with RFX 200 mg, respectively), and cardiovascular disease (18.8% vs. 25.2%, respectively). Finally, 35.0% of patients starting therapy with RFX 550 mg and 43.4% of those with RFX 200 mg had two or more extra-hepatic comorbidities.

In the 12 months following the start of antibiotic therapy, a significantly higher proportion of persistent patients (78.6% vs. 46.9%, *p* < 0.001) and a lower proportion of switchers (21.4% vs. 40.7%, *p* < 0.050) were found among the users of RFX 550 mg than RFX 200 mg (Table 3).

The multivariable logistic regression model (Table 4) revealed that the variables significantly associated with better persistence were older age (OR: 1.069; 95%CI: 1.027–1.113, *p* = 0.001), presence of portal hypertension (OR: 4.426; 95%CI: 1.278–15.335, *p* < 0.05), and RFX 550 mg as starting therapy after HE discharge (OR: 5.663; 95%CI: 2.389–133.423, *p* < 0.001). The *p*-value for the Hosmer–Lemeshow test was 0.2578.

As shown in Figure 2, the analysis of monthly doses (deaths and switches excluded) revealed that in patients started on RFX 550 mg, the mean monthly dose was 27,440 mg, in line with recommendations (1100 mg/day equivalent to ~33,000 mg/month), while in patients started on RFX 200 mg, the mean monthly dose was 11,629 mg, lower than the recommended dose (1200 mg/day equivalent to ~36,000 mg/month).

Then, the 447 patients hospitalized for HE and discharged alive were investigated according to the dispensing channel, DD, and DPC. Among them, 292 and 155 patients belonged to LHUs with DPC/DD and DD only, respectively (Figure 3). In regions with both distribution channels, 33.6% of patients received RFX 550 mg and 30.5% received RFX 200 mg, while in the regions with the DD channel only, 12.3% of patients received RFX 550 mg and 45.2% received RFX 200 mg.

The drug utilization profile was analyzed at 1-year follow-up among patients started on RFX 550 mg (deaths excluded): in the regions with DPC/DD, 81.4% of patients were persistent and 20.3% switched, whereas in the regions with DD only, 63.6% of patients were persistent and <4 patients switched (Figure 4).

The analysis of monthly doses (deaths and switches excluded) revealed that in patients receiving RFX in the regions with both channels for dispensing RFX 550 mg, the mean monthly dose was 27,523 mg for patients started on RFX 550 mg and 10,453 mg for patients started on RFX 200 mg, while in regions with DD delivery only, the mean monthly dose was 26,950 mg for patients started on RFX 550 mg and 12,400 mg for patients started on RFX 200 mg.

## 4. Discussion

The management of HE still remains challenging for clinicians and healthcare systems. The increased mortality associated with HE [17] was confirmed in this analysis where nearly 30% of patients died in-hospital during the first hospitalization for HE.

In this analysis, patients diagnosed with HE, approximately 40–45% were aged below 65 years. This relatively young age confirms the elevated clinical, economic, and social impact of HE attributable to both direct and indirect costs, as reported by recent US data [18]. In fact, besides the common requirement of caregivers and of hospital admission (often readmission), a further burden is represented by patients’ inability to work, despite their younger age, compared to other chronic conditions [18].

As expected, the patients with HE included had several hepatic and extra-hepatic comorbidities. Consistent with existing evidence, ascites and portal hypertension were the conditions most commonly found in the HE patients, as together these are the concomitant consequences of cirrhosis-induced liver dysfunction [19,20]. For many years, ammonia overload has been considered the main culprit of HE development in patients with cirrhosis [21]. However, some studies revealed that circulating ammonia levels do not always consistently correlate with the severity of HE, corroborating the complex pathophysiology of this disease and the involvement of other factors, above all pro-inflammatory status [22,23,24]. A systematic literature review and meta-analysis comparing data published up to June 2024 revealed that, besides a noticeable increase in blood ammonia, significant alterations were also found for other analytes in HE patients, including serum creatinine, albumin, sodium, and the pro-inflammatory cytokines interleukin-6 (IL-6) and tumor necrosis factor-alpha (TNFα) [25].

Given the poor prognosis of HE, timely and effective therapeutic interventions represent a landmark in reducing the risk of HE recurrence and re-hospitalizations for HE [3,4,26]. Here, we analyzed the characteristics of HE patients in a setting of real clinical practice in Italy, especially focusing on the rebounds of type of RFX form (550 mg or 200 mg) received by the patients in the 2-month period after HE-related hospital discharge on drug utilization and chances to achieve the recommended monthly medication dosage.

Among the treatment options for HE, RFX 550 mg in the formulation of two daily tablets is the only one authorized for reimbursement in Italy to prevent recurrent episodes of HE in adults [10]. However, many patients with HE are also prescribed RFX 200 mg (six tablets/day), although this formulation has different indications (specifically, acute and chronic intestinal infections sustained by Gram-positive and Gram-negative bacteria; diarrheal syndromes; diarrhea caused by altered balance of intestinal microbial flora; pre- and post-operative prophylaxis of infectious complications in gastrointestinal tract surgery; and adjuvant in the therapy of hyperammonemia). We have postulated that the use of RFX 200 mg might potentially impact drug utilization due to the higher number of pills to be taken daily and chronically to achieve disease remission. Indeed, we noticed that RFX 200 mg is prescribed more frequently than the licensed drug RFX 550 mg (in 35.6% and 26.2% of patients, respectively), suggesting suboptimal therapeutic management of HE. This drug utilization analysis emphasizes the need to improve RFX prescribing choices because RFX 550 mg is less used for treating HE patients, even though it was associated with higher persistence to treatment, lower switching rates, and an average monthly dose closer to recommendations over RFX 200 mg.

A recent real-world study in Southern Italy [14] showed that RFX treatment in HE was observed in the literature [27,28,29]; a substantial proportion of patients remain untreated, and among those who initiated treatment, only 35% were persistent after one year [14]. Broadly, the results of the present analysis and the available published data strongly endorse the paramount importance of optimizing HE therapeutic management, since a better use or a use in line with guidelines could improve persistence, reduce switching, achieve recommended dosing, and potentially improve patient outcomes [7,26,27,28,29].

To date, there are no similar analyses available focused on the DD/DPC distribution mix of RFX for the treatment of HE patients. In Italy, several drugs for the treatment of chronically ill patients are dispensed through the DD channel only [30]. Data from the Medicines Utilization Monitoring Centre (OsMed) on the consumption and expenditure of medicines supplied by the INHS [31] showed that access to medications by the patients is, in general, homogeneous throughout the territory; thus any observable differences between DPC and DD across the various regions could instead be attributable to different epidemiological and demographic profiles, as well as different levels of appropriate drug prescribing (defined as indication, frequency, or dose of the medication in line with recommendations) [32,33]. In Italy, RFX 550 mg is dispensed either by the hospital (DD) or via territorial pharmacies (DPC), with some differences according to local directives, given that some regions have only the DD channel, and others have both DD and DPC. We postulated that these different modalities might affect the prescription appropriateness and drug utilization of RFX in the setting of HE. In this context, some important points emerged from our analysis. First, in the regions where RFX 550 mg is also distributed through the community pharmacies (DPC), greater prescriptive appropriateness was observed since the proportion of patients who received the licensed drug was higher than those prescribed with RFX 200 mg. In addition, easier access to RFX 550 mg using the channel of territorial pharmacies improved persistence, as 81.4% of patients in the regions with DPC/DD delivery remained on therapy with RFX 550 mg, compared with 63.6% for the regions with DD only. Taken together, these data suggest that making the licensed drug available to the patients via both dispensing routes, hospital and community pharmacies, might improve prescription appropriateness and therapy compliance in terms of adherence and persistence. Thus, the results of the present analysis have the potential to provide useful insights for future healthcare policies aimed at optimizing the distribution of medications, drug utilization, and prescription behaviors.

This retrospective observational analysis has strengths and limitations to be recognized. To the best of our knowledge, this is the first comparative investigation focused on the two available formulations of RFX in consideration of drug utilization and the type of distribution of RFX 550 mg, via hospital or community pharmacies. On the other hand, the use of administrative databases might result in missing or incomplete data. For instance, comorbidities and cirrhosis-related hepatic complications were detected using hospitalization codes or drug prescriptions as a diagnosis proxy, so disease severity could not be evaluated. Another flaw lies in the small sample size of some subgroups when analyzing drug utilization profiles according to the drug dispensation channels adopted by the included LHUs. Lastly, this is an analysis meant to describe the current management of HE in Italy, so given that the type of RFX preparations available abroad, dispensation models, and prescription modalities are different in other countries, the results cannot be generalized on a larger international scale.

## 5. Conclusions

The present findings, which emerged from real clinical practice in Italy, suggest that the treatment of patients with HE with RFX might be optimized by focusing on some still inadequate prescribing choices. Even though RFX 550 mg is the only formulation of the antibiotic with specific indication and reimbursability for the prevention of recurrent episodes of HE, RFX 200 mg is also used in a significant proportion of patients, even higher than that of patients with rifaximin 550 mg (35.6% and 26.2%, respectively). The analysis also revealed that the possibility in some regions to distribute RFX 550 mg either through the hospital route or via the territorial pharmacies led to a greater prescriptive appropriateness because a larger proportion of patients received the licensed formulation when both channels were available.

## Figures and Tables

**Figure 1 medicina-61-00221-f001:**
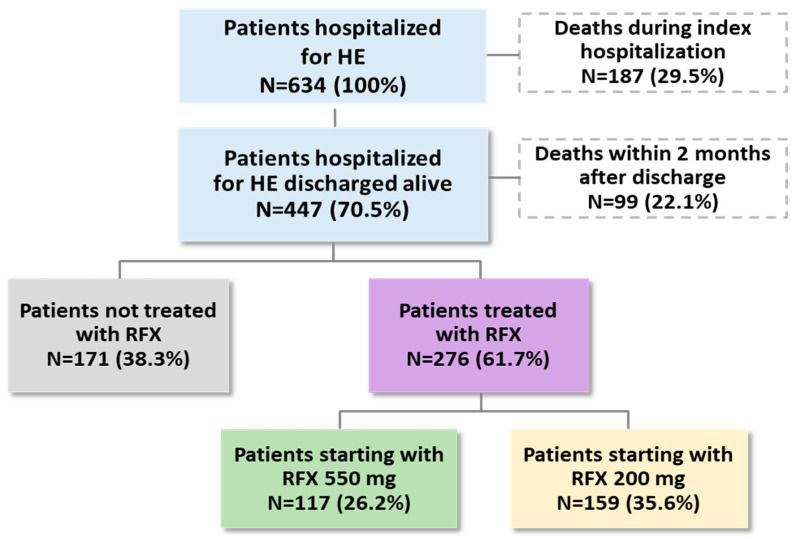
Flowchart of patients’ selection: presence of treatment and deaths during the first 2 months after discharge from index hospitalization. Abbreviations: HE, hepatic encephalopathy; RFX, rifaximin-α.

**Figure 2 medicina-61-00221-f002:**
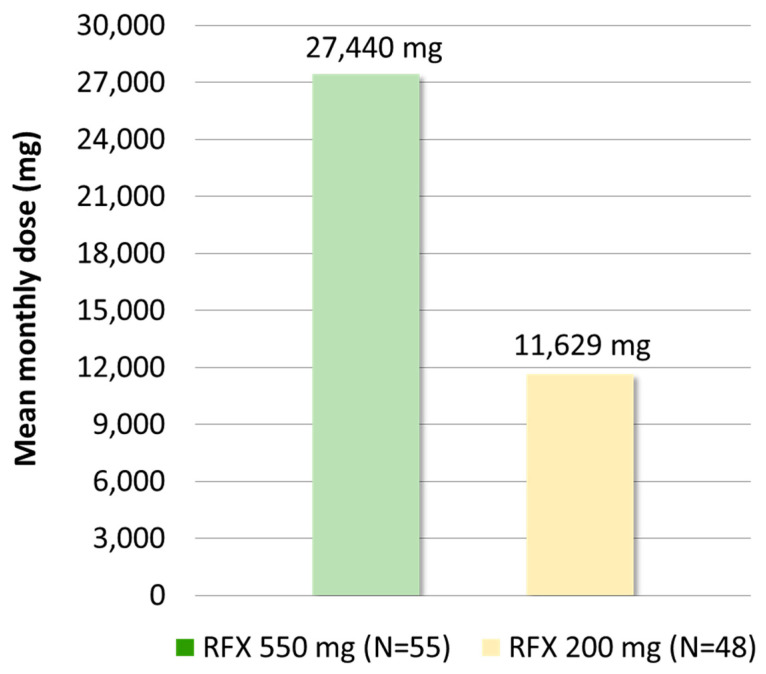
Mean monthly dose in mg (deaths and switches excluded) in patients started on RFX 550 mg or with RFX 200 mg.

**Figure 3 medicina-61-00221-f003:**
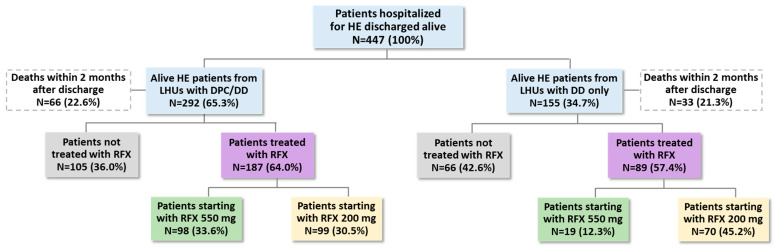
Presence of RFX treatment, first 2 months after discharge for HE, by type of delivery, DPC and DD, or DD only available in the different regions. Abbreviations: DD, direct distribution; DPC, “distribuzione per conto” (on-behalf distribution); HE, hepatic encephalopathy; LHUs, local health units; RFX, rifaximin-α.

**Figure 4 medicina-61-00221-f004:**
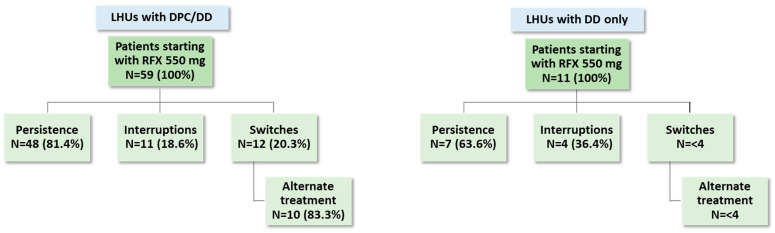
Drug utilization profiles in terms of persistence, interruptions, switches, and alternate treatment among HE patients stratified by type of dispensation model adopted in their LHU. Abbreviations: DD, direct distribution; DPC, “distribuzione per conto” (on-behalf distribution); RFX, rifaximin-α.

**Table 1 medicina-61-00221-t001:** Main differences between RFX 550 mg and 200 mg formulations.

	Rifaximin 550 mg	Rifaximin 200 mg
Indications	■ Reduction in recurrences of episodes of overt HE in patients aged ≥ 18 years.	(1) Acute and chronic intestinal infections sustained by Gram+ and Gram− bacteria; diarrheic syndrome. (2) Diarrhea caused by an altered equilibrium of the intestinal microbial flora (summer diarrhea, traveler’s diarrhea, enterocolitis). (3) Peri-operative prophylaxis of infective complications in surgery of the gastroenteric tract. (4) Adjuvant in the treatment of hyperammonemia.
Administration route	Oral.	Oral.
Form	Film-coated tablets.	■ Film-coated tablets. ■ Granules for oral suspension.
Recommended daily dose	Adults: 2 daily tablets 1100 mg.	Different according to the indications. For the indication (4) in adults and children over 12 years: 2 tablets of 200 mg or 20 mL oral suspension (equivalent to 400 mg RFX) every 8 h. At the doctor’s discretion, the dosage may be varied in quantity and frequency. Unless otherwise prescribed, treatment should not exceed 7 days.
Clinical efficacy	The efficacy and safety of RFX administered at a dose of 550 mg twice daily to adult patients in remission from HE were described in the 6-month phase 3, randomized, double-blind, placebo-controlled trial RFHE3001 [4].	Clinical studies in patients with traveler’s diarrhea have demonstrated the clinical efficacy of RFX 200 mg against ETEC (*Enterotoxigenic E. coli*) and EAEC (*Enteroaggregative E. coli*). These bacteria are mainly responsible for traveler’s diarrhea in individuals traveling to Mediterranean countries or tropical and subtropical regions [9].
Prescription mode in Italy	Medicinal products subject to restrictive medical prescriptions (dispensed to patients on prescription by hospitals or by gastroenterologists, infective disease specialists, or internists).	Reimbursable essential drugs and drugs for chronic diseases that can be prescribed by the general practitioner and are dispensed by community pharmacies.
Distribution in Italy	Direct distribution (hospital) or distribution via community pharmacies on behalf of LHUs.	Distribution through community pharmacies.

**Table 2 medicina-61-00221-t002:** Baseline demographic and clinical characteristics of the patients stratified by type of starting RFX treatment.

	Patients Started on RFX 550 mg (N = 117)	Patients Started on RFX 200 mg (N = 159)	*p*
Male gender, N (%)	65 (55.6%)	104 (65.4%)	0.097
Age at inclusion, mean (±SD)	65.9 (±11.1)	67.3 (±11.4)	0.315
Age classes			
<65 years, N (%)	53 (45.3%)	64 (40.3%)	0.691
65–74 years, N (%)	34 (29.1%)	49 (30.8%)	
≥75 years, N (%)	30 (25.6%)	46 (28.9%)	
Cirrhosis complications			
Ascites, N (%)	51 (43.6%)	55 (34.6%)	0.129
Portal hypertension, N (%)	23 (19.7%)	28 (17.6%)	0.665
Liver primary malignant tumors, N (%)	11 (9.4%)	14 (8.8%)	0.864
Alcohol dependence syndrome, N (%)	<4	4 (2.5%)	0.650
Extra-hepatic comorbidities			
Diabetes, N (%)	57 (48.7%)	80 (50.3%)	0.793
Cancer, N (%)	24 (20.5%)	41 (25.8%)	0.308
Cardiovascular disease, N (%)	22 (18.8%)	40 (25.2%)	0.211
Psychiatric conditions, N (%)	18 (15.4%)	16 (10.1%)	0.184
Cerebrovascular disease, N (%)	14 (12.0%)	26 (16.4%)	0.306
Chronic kidney disease, N (%)	13 (11.1%)	13 (8.2%)	0.409
Number of extra-hepatic comorbidities			
0, N (%)	32 (27.4%)	37 (23.3%)	0.372
1, N (%)	44 (37.6%)	53 (33.3%)	
≥2, N (%)	41 (35.0%)	69 (43.4%)	

**Table 3 medicina-61-00221-t003:** Drug utilization and outcomes at 12-month follow-up among patients starting treatment with RFX 550 mg and RFX 200 mg (deaths excluded).

	Patients Started on RFX 550 mg, Alive (N = 70)	Patients Started on RFX 200 mg, Alive (N = 81)	*p*
Drug utilization			
Persistence, N (%)	55 (78.6%)	38 (46.9%)	**<0.001**
Interruptions, N (%)	15 (21.4%)	43 (53.1%)	**<0.001**
Switches, N (%)	15 (21.4%) *	33 (40.7%) *	**<0.05**

* Of 15 switchers with RFX 550 mg, 13 (86.7%) alternated treatments, and of 33 switchers with RFX 200 mg, 5 (15.2%) alternated treatments. Significant *p*-values are highlighted in bold.

**Table 4 medicina-61-00221-t004:** Logistic regression model for predictors of persistence.

	OR	95%CI		*p*
Age	1.069	1.027	1.113	**0.001**
Gender (ref. female)	1.522	0.659	3.519	0.326
Ascites (ref. absence)	1.545	0.606	3.939	0.362
Portal hypertension (ref. absence)	4.426	1.278	15.334	**<0.050**
Liver primary malignant tumors (ref. absence)	0.485	0.097	2.431	0.379
Diabetes (ref. absence)	1.039	0.468	2.307	0.924
Cardiovascular disease (ref. absence)	1.829	0.572	5.850	0.309
Cerebrovascular disease (ref. absence)	1.389	0.393	4.906	0.610
Psychiatric conditions (ref. absence)	0.896	0.314	2.554	0.837
Started treatment				
RFX 200 mg (ref.)	1.000			
RFX 550 mg	5.663	2.389	13.423	**<0.001**

Significant *p*-values are highlighted in bold. Abbreviations: CI, confidence interval; OR, odds ratio.

## Data Availability

All data used for the current study are available upon reasonable request to CliCon S.r.l. Società Benefit, which is the body entitled to data treatment and analysis by local health units.

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
