# Peer review of "Drug Utilization of Rifaximin-α in Patients with Hepatic Encephalopathy: Evidence from Real Clinical Practice in Italy"

_medicina, 2025, doi:10.3390/medicina61020221_

Round 1
Reviewer 1 Report
Comments and Suggestions for Authors
Review
Dear Editor,
Dear Authors,
I have read the manuscript titled „Drug Utilization of Rifaximin-α in Patients with Hepatic Encephalopathy: Evidence from the Real Clinical Practice in Italy” with great interest. The primary aim of this retrospective, observational, registry-based study was to investigate whether rifaximin formulation (550 mg or 200 80 mg) prescribed to patients with hepatic encephalopathy and its distribution modality influences patient compliance. The authors have shown that the appropriate formulation of RFX is not prescribed enough for its indication, and that this formulation has beneficial effects on patient adherence. I believe the idea and objective of the study are very interesting, and that patients could greatly benefit from these observations in real life. However, I do have several remarks, please see the comments below. Additionally, before further processing, the article needs thorough English check, since my understanding of the article was limited in several paragraphs due to poor English.
Line 17: Please do not use abrreviations.
Line 22: Please specify dosing regimen.
Line 49: Please delete „recurrence“.
Lines 50-52: Please rephrase. This varies by regions or countries – you could specify what is it like in your country. Additionally, the second part of sentence is not clear.
Lines 54-55: Could you please specify the reason behind this clinical practice?
Line 69: Please define „hospital channel“. Does DD system of drug distribution means that patients get medications directly from the hospital they were treated in? Could you please be more specific, I failed to understand the exact differences between the two systems, or their advantages and disadvantages – e.g. which one is more convinient for patients and their caregivers, how do patients get selected for one system or the other, etc...
Line 74: Please rephrase.
Line 75: Does this mean that RFX 550 mg is not reimbursable?
Lines 93-98: If you are using roman numerals, they should be in capital.
Line 103: Please define „aggregated form“.
Figure 1. Please re-check the Figure 1 since it is not quite clear. The sum of untreated, not treated with RFX, treated with RFX and patients who died, does not equal the number of patients discharged alive. Moreover, the sum of these categories is 664, which is more than the total number of hospitalized patients for HE during the study period. Please correct this and provide more understantable flow-chart.
Line 205: Please use the abrreviations you have already introduced.
Table 1. Please delete „abrreviations“ in the table footnote.
Line 240: Could you please specify which patients are there? Are these patients the ones with HE, or have you included all patients who were prescribed with RFX during the study period? If that is the case, I would advise excluding this part of the analyses, given that your intervention is intended only for the specific subgroup of patients – those with end-stage liver disease and HE.
Figure 4. I advise making this Figure more understadable. The proportion of patients that have died (labeled as „deaths“), should be presented on the side, as the figure branch, since untreated patients, plus pts terated with RFX and those who died do not sum up to 292, and therefore cannot be presented as proportion of that subgroup.
Figure 5. The results need more consistent presentation.
Line 367: The conclusion needs to be rephrased in order to briefly summarize the most significant results.
Comments on the Quality of English LanguageThe manuscript would greatly benefit from English editing, given that manuscript understanding is limited in this form
Author Response
COMMENT 1. I have read the manuscript titled “Drug Utilization of Rifaximin-α in Patients with Hepatic Encephalopathy: Evidence from the Real Clinical Practice in Italy” with great interest. The primary aim of this retrospective, observational, registry-based study was to investigate whether rifaximin formulation (550 mg or 200 80 mg) prescribed to patients with hepatic encephalopathy and its distribution modality influences patient compliance. The authors have shown that the appropriate formulation of RFX is not prescribed enough for its indication, and that this formulation has beneficial effects on patient adherence. I believe the idea and objective of the study are very interesting, and that patients could greatly benefit from these observations in real life. However, I do have several remarks, please see the comments below. Additionally, before further processing, the article needs thorough English check, since my understanding of the article was limited in several paragraphs due to poor English.
AUTHORS’ REPLY 1: Dear reviewer, we have now extensively proofread the manuscript to improve English and understandability.
COMMENT 2. Line 17: Please do not use abrreviations.
AUTHORS’ REPLY 2: Dear reviewer, we would like to address your concern, but the word limit for the abstract is very strict (300 words), this is the reason why we used abbreviations. However, the journal allows their use of in the abstract (please see this very recent publication at the link: https://www.mdpi.com/1648-9144/61/1/91)
COMMENT 3. Line 22: Please specify dosing regimen.
AUTHORS’ REPLY 3: We have now added the recommended daily dosage for each Rifaximin formulation, 550 mg and 200 mg.
COMMENT 4. Line 49: Please delete „recurrence“.
AUTHORS’ REPLY 4: We have rephrased, reporting what it is stated in the EASL guidelines (ref. 6).
COMMENT 5. Lines 50-52: Please rephrase. This varies by regions or countries – you could specify what is it like in your country. Additionally, the second part of sentence is not clear.
AUTHORS’ REPLY 5: We have extensively rephrased and reorganized this part.
COMMENT 6. Lines 54-55: Could you please specify the reason behind this clinical practice?
AUTHORS’ REPLY 6: We have substantially reorganized the introduction because your question is indeed the main research question of this analysis. Based on our results, one main reason behind the attitude to use the RFX formulation of 200 mg rather than the one suggested by the guidelines and reimbursed to reduce the risk of overt HE recurrence in adults (RFX 550 mg) lies in the easier access to the medication for the patients, because it can be prescribed by the GP and acquired by the patients in the territorial pharmacies.
COMMENT 7. Line 69: Please define „hospital channel“. Does DD system of drug distribution means that patients get medications directly from the hospital they were treated in? Could you please be more specific, I failed to understand the exact differences between the two systems, or their advantages and disadvantages – e.g. which one is more convenient for patients and their caregivers, how do patients get selected for one system or the other, etc...
AUTHORS’ REPLY 7: Dear reviewer, to better explain the difference between the two distribution models, we have now added some more details to the text. The direct distribution means that the patients get their medication at the hospital pharmaceutical unit, but this does not imply that the patients are necessarily hospitalized.
COMMENT 8. Line 74: Please rephrase.
COMMENT 9. Line 75: Does this mean that RFX 550 mg is not reimbursable?
AUTHORS’ REPLY 8-9: We have rephrased in order to make this section clearer
COMMENT 10. Lines 93-98: If you are using roman numerals, they should be in capital.
AUTHORS’ REPLY 10: Roman numerals used as numbered list are commonly used in lowercase.
COMMENT 11. Line 103: Please define „aggregated form“.
AUTHORS’ REPLY 11: We have rephrased and simplified this section (lines 129-130).
COMMENT 12. Figure 1. Please re-check the Figure 1 since it is not quite clear. The sum of untreated, not treated with RFX, treated with RFX and patients who died, does not equal the number of patients discharged alive. Moreover, the sum of these categories is 664, which is more than the total number of hospitalized patients for HE during the study period. Please correct this and provide more understantable flow-chart.
AUTHORS’ REPLY 12: This has been done, modifying both text and figure.
COMMENT 13. Line 205: Please use the abrreviations you have already introduced.
AUTHORS’ REPLY 13: Since they were only mentioned once, we preferred removing the abbreviations CKD and CVD in the methods (line 164), if you agree.
COMMENT 14. Table 1. Please delete „abrreviations“ in the table footnote.
AUTHORS’ REPLY 14: This has been done.
COMMENT 15. Line 240: Could you please specify which patients are there? Are these patients the ones with HE, or have you included all patients who were prescribed with RFX during the study period? If that is the case, I would advise excluding this part of the analyses, given that your intervention is intended only for the specific subgroup of patients – those with end-stage liver disease and HE.
AUTHORS’ REPLY 15: This part of the analysis was meant to investigate drug utilization stratifying the users of RFX by naïve and experienced. But this section might be removed since the main focus of the analysis in RFX treatment in HE patients, as you properly noticed.
COMMENT 16. Figure 4. I advise making this Figure more understadable. The proportion of patients that have died (labeled as „deaths“), should be presented on the side, as the figure branch, since untreated patients, plus pts terated with RFX and those who died do not sum up to 292, and therefore cannot be presented as proportion of that subgroup.
AUTHORS’ REPLY 16: This has been done, modifying both text and figure.
COMMENT 17. Figure 5. The results need more consistent presentation.
AUTHORS’ REPLY 17: The results described in the text are consistent with the figure. We only avoided to mention the rate of interruptions because it complementary to the rate of persistence (i.e. the sum is 100%). Anyhow, the figure has been slightly modified to help its readability.
COMMENT 18. Line 367: The conclusion needs to be rephrased in order to briefly summarize the most significant results.
AUTHORS’ REPLY 18: This has been done.
COMMENT 19. Comments on the Quality of English Language
The manuscript would greatly benefit from English editing, given that manuscript understanding is limited in this form
AUTHORS’ REPLY 19: The manuscript has been now extensively proofread for English Language. Dear reviewer, we would like to thank you for these valuable suggestions that allowed us to improve the clarity and hopefully the scientific soundness of our work.
Reviewer 2 Report
Comments and Suggestions for Authors
This manuscript provides a detailed real-world analysis of rifaximin utilization in Italian patients with hepatic encephalopathy (HE). The study compares the prescribing patterns, persistence rates, and outcomes between two formulations of rifaximin (550 mg and 200 mg) and explores the impact of distribution channels (direct distribution vs. community pharmacy) on treatment adherence. Using retrospective data from Italian healthcare entities, the authors highlight suboptimal prescribing practices and emphasize the importance of aligning treatment with clinical guidelines to optimize HE management.
Please consider the following suggestions:
1. Consider defining "persistence" in the abstract. While it is well-defined under the methods section, someone reading only the abstract might misinterpret this outcome.
2. Clarify in the abstract that the 637 HE patients included in the study were hospitalized (not outpatient), as this distinction is crucial for context.
3. Elaborate on what "mean monthly dose closer to recommendations" means in the abstract. Specify that this refers to the alignment of the prescribed dose with clinical guidelines for effective HE management.
4. Provide details on whether the logistic regression model used is a bivariate or multivariate analysis. If it is multivariate, elaborate on the goodness-of-fit metrics (e.g., Hosmer-Lemeshow test or pseudo R² values) to assess the model's validity. If it is bivariate, consider conducting a multivariate analysis to adjust for potential confounding variables and enhance the robustness of the findings.
5. Expand on the statement: "Providing RFX 550 mg solely via the channel of hospital distribution can have negative repercussions in terms of appropriateness and persistence to treatment." Tie this insight more directly to the study's conclusions, emphasizing how distribution channels influence treatment outcomes.
6. Provide comments on future directions, such as policy changes to optimize drug distribution, strategies to improve adherence, or further studies to assess patient-reported outcomes and quality of life.
This manuscript is a valuable contribution to the field of HE management and highlights important gaps in rifaximin utilization. Addressing the suggested areas for improvement will enhance its impact and practical relevance, particularly for policymakers and healthcare providers. I recommend the manuscript for publication after minor revisions to address the points outlined above.
Author Response
Comments and Suggestions for Authors
This manuscript provides a detailed real-world analysis of rifaximin utilization in Italian patients with hepatic encephalopathy (HE). The study compares the prescribing patterns, persistence rates, and outcomes between two formulations of rifaximin (550 mg and 200 mg) and explores the impact of distribution channels (direct distribution vs. community pharmacy) on treatment adherence. Using retrospective data from Italian healthcare entities, the authors highlight suboptimal prescribing practices and emphasize the importance of aligning treatment with clinical guidelines to optimize HE management.
Please consider the following suggestions:
COMMENT 1. Consider defining "persistence" in the abstract. While it is well-defined under the methods section, someone reading only the abstract might misinterpret this outcome.
AUTHORS’ REPLY 1: This is a reasonable concern, we shortly defined the meaning of “persistent patients” in brackets, since the journal rules allow only 300 words in the abstract.
COMMENT 2. Clarify in the abstract that the 637 HE patients included in the study were hospitalized (not outpatient), as this distinction is crucial for context.
AUTHORS’ REPLY 2: This has been done.
COMMENT 3. Elaborate on what "mean monthly dose closer to recommendations" means in the abstract. Specify that this refers to the alignment of the prescribed dose with clinical guidelines for effective HE management.
AUTHORS’ REPLY 3: This has been done.
COMMENT 4. Provide details on whether the logistic regression model used is a bivariate or multivariate analysis. If it is multivariate, elaborate on the goodness-of-fit metrics (e.g., Hosmer-Lemeshow test or pseudo R² values) to assess the model's validity. If it is bivariate, consider conducting a multivariate analysis to adjust for potential confounding variables and enhance the robustness of the findings.
AUTHORS’ REPLY 4: We have now added more details to the section “Statistical analysis” (lines 202-203) and in the results. The logistic model is a multivariate analysis. The p-value resulted using the Hosmer-Lemeshow test has been included in the results (line 248).
COMMENT 5. Expand on the statement: "Providing RFX 550 mg solely via the channel of hospital distribution can have negative repercussions in terms of appropriateness and persistence to treatment." Tie this insight more directly to the study's conclusions, emphasizing how distribution channels influence treatment outcomes.
- Provide comments on future directions, such as policy changes to optimize drug distribution, strategies to improve adherence, or further studies to assess patient-reported outcomes and quality of life.
AUTHORS’ REPLY 5: Thanks for these valuable suggestions that allowed us to better elucidate the points you mentioned (lines 364-369).
COMMENT 6. This manuscript is a valuable contribution to the field of HE management and highlights important gaps in rifaximin utilization. Addressing the suggested areas for improvement will enhance its impact and practical relevance, particularly for policymakers and healthcare providers. I recommend the manuscript for publication after minor revisions to address the points outlined above.
AUTHORS’ REPLY 6: Dear reviewer, we sincerely thank you for the comments and the overall positive feedback of our work.
Reviewer 3 Report
Comments and Suggestions for Authors
This is a retrospective observational study of real-world drug use, addressing a topic of low prevalence but high health impact. Prescription and its characteristics are evaluated with respect to effectiveness and strategies to improve it, enabling the reduction of complications and costs.
The title is clear, synthetic and very descriptive of the type of study and topic.
The abstract provides a clear summary of the main concepts of each section, providing objective, quantified results without significant bias.
The introduction is brief, synthetic and appropriate. It provides the main concepts related to the topic and justifies the research questions, while adequately presenting the objectives.
The materials and methods are adequately described, allowing for the reproducibility of the study. However, it would be good to improve the description of the inclusion and exclusion criteria, which are described in a general way. It is advisable to state them clearly and separately for each one of them.
It would be advisable to describe in a small table the differences of the rifaximin formulations analyzed. The results are described in a clear, concrete, objective manner, with appropriate units and using adequate statistical tests. The tables are very well presented and are correct. The figures and graphs use colors that do not seem the most appropriate for visualization, offering a strident contrast. It is suggested that they be changed to softer tones. In figure 2, it is necessary to include lines that allow the magnitude of the proportions to be quantified.
The discussion is quite adequate. It is compared with the bibliography on the subject, with other similar results, and the potential causes and benefits are discussed. The limitations should be expanded, for example: source of the data, ethnic characteristics, difficulty in extrapolating to other populations, differences with preparations available internationally, possibility of blinding in the evaluation, need for subsequent prospective studies, etc.
Author Response
This is a retrospective observational study of real-world drug use, addressing a topic of low prevalence but high health impact. Prescription and its characteristics are evaluated with respect to effectiveness and strategies to improve it, enabling the reduction of complications and costs.
The title is clear, synthetic and very descriptive of the type of study and topic.
The abstract provides a clear summary of the main concepts of each section, providing objective, quantified results without significant bias.
The introduction is brief, synthetic and appropriate. It provides the main concepts related to the topic and justifies the research questions, while adequately presenting the objectives.
The materials and methods are adequately described, allowing for the reproducibility of the study. COMMENT 1. However, it would be good to improve the description of the inclusion and exclusion criteria, which are described in a general way. It is advisable to state them clearly and separately for each one of them.
AUTHORS’ REPLY 1: This point has been addressed (lines 133-135 and lines 140-141).
COMMENT 2. It would be advisable to describe in a small table the differences of the rifaximin formulations analyzed.
AUTHORS’ REPLY 2: Dear reviewer, we have added a table (current Table 1 at page 3) to describe the differences between the two rifaximin formulations analyzed.
The results are described in a clear, concrete, objective manner, with appropriate units and using adequate statistical tests. The tables are very well presented and are correct.
COMMENT 3. The figures and graphs use colors that do not seem the most appropriate for visualization, offering a strident contrast. It is suggested that they be changed to softer tones. In figure 2, it is necessary to include lines that allow the magnitude of the proportions to be quantified.
AUTHORS’ REPLY 3: This has been done. We have used softer tones in the figure as you suggested and included the lines to the plot in fig. 2.
The discussion is quite adequate. It is compared with the bibliography on the subject, with other similar results, and the potential causes and benefits are discussed.
COMMENT 4. The limitations should be expanded, for example: source of the data, ethnic characteristics, difficulty in extrapolating to other populations, differences with preparations available internationally, possibility of blinding in the evaluation, need for subsequent prospective studies, etc.
AUTHORS’ REPLY 4: Along with your proper advice, we have now added some additional points to the limitations. Dear reviewer, we did our best to address all the points raised, where possible. The authors sincerely thank you for the comments and the overall positive feedback of our work.